# Anthrax, livelihood vulnerability, and food insecurity in selected game management areas in Zambia: A mixed-methods analysis at the human-wildlife-livestock interface

Laila Gondwe[1], Chisoni Mumba[1]*, Kezzy Besa[1], Davies Phiri[2], Exillia Kabbudula[1], Noanga Mebelo[1], Suwilanji S. Sichone[3], Ntazana N. Sinyangwe[1], Mwila Kayula[4], Geoffrey Mainda[4], Fredrick M. Kivaria[5], Charles Bebay[5], Baba Soumare[6], Mtui-Malamsha J. Niwael[4], Chitwambi Makungu[4]

**1** Department of Disease Control, School of Veterinary Medicine, University of Zambia, Lusaka, Zambia, **2** Department of Community Education and Lifelong Learning, University of Zambia, Lusaka, Zambia, **3** Pangolin Protection, Wild Crime Prevention (WCP), Chilanga, Zambia, **4** Emergency Centre for Transboundary Animal Diseases (ECTAD), The Food and Agriculture Organization of the United Nations (FAO), Lusaka, Zambia, **5** Emergency Centre for Transboundary Animal Diseases (ECTAD), The Food and Agriculture Organization of the United Nations (FAO), Nairobi, Kenya, **6** Emergency Centre for Transboundary Animal Diseases (ECTAD), The Food and Agriculture Organization of the United Nations (FAO), Rome, Italy

* sulemumba@yahoo.com, cmumba@unza.zm

## Abstract

The study aimed to investigate the socio-economic impacts of anthrax outbreaks on rural communities in selected Game Management Areas (GMAs) of Zambia, with a particular focus on how livelihood diversification influences exposure to zoonotic disease risk. We used a mixed-methods approach to assess how environmental, economic, and social factors interact to shape community vulnerability and resilience. The central hypothesis is that proximity to wildlife and reliance on high-risk alternative livelihoods, such as charcoal burning, fishing, and unregulated game meat consumption, heighten household exposure to anthrax, particularly in contexts of limited veterinary access and social protection. Quantitative results showed that 87.9% of households were adversely affected by drought, with 69.1% receiving no external assistance. Coping strategies included charcoal burning, fishing, and gardening, with less than 30% benefiting from social cash transfer programs. Qualitative data provided depth to these findings, illustrating how anthrax outbreaks compounded food insecurity by decimating livestock, a critical source of food and income. Participants emphasized the need for livelihood diversification, such as beekeeping, poultry farming, and small-scale businesses, to mitigate the dual challenges of climatic shocks and zoonotic diseases. Gendered dimensions of vulnerability were also evident, with women disproportionately affected by food insecurity and economic barriers. This integration of quantitative and qualitative data highlights the complex interplay of environmental, economic, and social factors influencing resilience in rural

**Data availability statement:** All relevant data are in the manuscript.

**Funding:** The authors received no specific funding for this work.

**Competing interests:** The authors have declared that no competing interests exist.

communities. The study underscores the urgent need for comprehensive interventions to address systemic vulnerabilities, promote livelihood diversification, and strengthen food security. Tailored approaches, particularly those empowering women and marginalized groups, are crucial for enhancing community resilience and reducing the impacts of zoonotic diseases like anthrax.

## Author summary

Anthrax is a recurring zoonotic disease in Zambia, particularly in Game Management Areas (GMAs), where human-wildlife-livestock interactions are high. This study examines the socio-economic impacts of anthrax outbreaks on rural communities, focusing on food insecurity and livelihood vulnerabilities. Using a mixed-methods approach, we collected quantitative data from structured surveys and qualitative insights from focus group discussions across three study sites: Simalaha Community Conservancy, Lochinvar, and Blue Lagoon National Parks. Our findings highlight that most households depend on subsistence farming and livestock rearing as their primary livelihoods. However, frequent droughts, limited access to education, and inadequate social safety nets exacerbate economic hardship. The study reveals that anthrax outbreaks significantly disrupt food security by decimating livestock, an essential source of nutrition and income. Communities adopt alternative coping strategies such as charcoal burning, fishing, and gardening, yet these activities may inadvertently increase exposure to anthrax spores and other zoonotic risks. This study underscores the urgent need for integrated interventions to enhance community resilience. Policies promoting livelihood diversification, such as beekeeping and poultry farming, could help mitigate the dual threats of climatic shocks and zoonotic diseases. Moreover, gender-sensitive approaches are essential, as women bear a disproportionate burden of food insecurity. Strengthening disease surveillance, improving access to veterinary services, and supporting sustainable economic activities are crucial steps toward safeguarding both public health and rural livelihoods in GMAs in Zambia.

## 1. Introduction

Game Management Areas (GMAs) aim to balance wildlife conservation with regulated human activities, presenting unique opportunities and challenges for the communities residing within them [1]. In Africa, including Zambia, livestock farming is a cornerstone of rural livelihoods, providing income, food security, and ecosystem services such as grazing, firewood, and medicinal plants [2,3]. However, livestock farmers in GMAs face mounting pressures from environmental changes, economic constraints, and the need to diversify income sources.

In response, many farmers have adopted alternative livelihood strategies such as crop farming, fishing, small-scale businesses, and wood collection [1]. While these activities enhance financial stability, they also increase the risk of zoonotic disease

transmission such as anthrax. Anthrax, caused by *Bacillus anthracis*, is particularly concerning due to its persistence in soil and its transmission through contact with infected animals or contaminated environments [4]. In GMAs, the intersection of human and wildlife habitats amplifies anthrax exposure risks [5,6]. This is further exacerbated by climate change, which drives migration in search of pasture and food, often resulting in encroachment into protected wildlife areas [7,8]. Such encroachment increases human-wildlife interactions and conflicts, biodiversity loss, and zoonotic disease transmission. Improper disposal of infected carcasses during outbreaks compounds environmental contamination and infection cycles.

Recent outbreaks highlight the endemic nature of anthrax in Zambia. In 2023, an outbreak in Sinazongwe District, Southern Province, led to the deaths of two hippos and four cattle [9]. This region, heavily reliant on livestock farming, has experienced repeated outbreaks, underscoring the need for enhanced disease surveillance and mitigation strategies.

This study examines how livelihood diversification influences anthrax exposure among livestock farmers in GMAs, highlighting the interconnected risks for animals and humans. We investigate socio-economic vulnerabilities that increase human exposure, including financial constraints limiting livestock vaccination, reliance on high-risk meat consumption, and occupational hazards faced by farmers handling infected carcasses. Furthermore, we assess how human-wildlife interactions shape livestock management, particularly whether increased contact with wildlife alters grazing behaviors, expands disease transmission pathways, or affects farmers' disease mitigation decisions. The underlying hypothesis is that proximity to wildlife and shared environmental resources heightens anthrax risk by increasing livestock exposure to contaminated soils and carcasses, which in turn raises human infection risk. We integrate ecological, epidemiological, and socio-economic perspectives to inform strategies that promote sustainable livelihoods, wildlife conservation, and public health protection.

## 2. Methods

### 2.1. Ethics statement

We obtained ethical clearance from Excellence in Research Ethics and Science (ERES) Converge, reference number "2024-Aug-015." We anonymized names of participants of Focus group discussions in the raw narratives using pseudonyms (A1, A2, A…) to safeguard confidentiality. We also de-identified all participant responses during transcription and securely stored raw data in compliance with ethical guidelines. We fully informed participants about the objectives of the study and assured them that their participation was voluntary, with the option to withdraw without any negative consequences. Formal verbal consent was obtained from participants. All research procedures complied with applicable ethical and regulatory guidelines for human subjects' research in Zambia.

### 2.2. Study design

We employed a mixed-method research design, combining qualitative and quantitative data collection approaches. Focus groups were selected as a crucial qualitative technique to gather group perspectives, investigate socioeconomic factors affecting food security, knowledge, attitudes, and behaviours related to anthrax exposure, and promote candid conversations about the difficulties in managing the disease in GMAs. This method ensured a contextually sensitive and participative investigation of anthrax risk while enabling a comparative study of various livelihood choices. For quantitative data, we used a structured questionnaire.

### 2.3. Description of the study sites

We conducted the study in Simalaha Community Conservancy, Lochinvar, and Blue Lagoon National Parks in Southern and Western Provinces of Zambia, as shown in Fig 1. Lochinvar and Blue Lagoon National Parks were selected because of the recent anthrax outbreaks in Southern Province which originated from wildlife (hippos). Simalaha Community

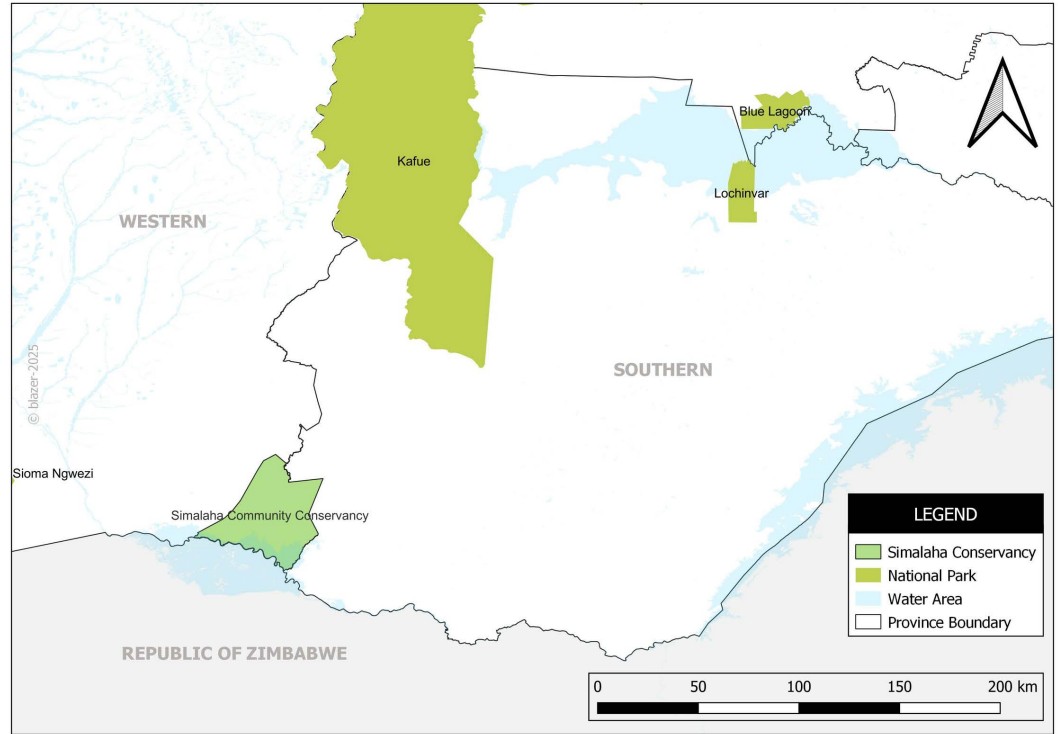

**Fig 1. Map showing the study sites.** Developed by authors. Reprinted from [Zambia_Mosaic_250Karc1950_ddecw] under a CC BY licence, with permission from the Surveyor General, Government Republic of Zambia, original copyright, [1974].

Conservancy was selected because of the endemicity of anthrax disease in Western Province. These study areas also host the largest livestock population in Zambia, a critical human-wildlife-livestock interface area which makes understanding of disease dynamics better.

Simalaha Conservancy is located in Western Province of Zambia. This conservancy encompasses communities from Mwandi and parts of Sesheke districts. Simalaha plays a critical role in wildlife conservation and community livelihoods, as it lies within one of the six key wildlife dispersal corridors of the Kavango-Zambezi Transfrontier Conservation Area (KAZA TFCA) [10]. Specifically, it is part of the Chobe-Zambezi dispersal area, which connects Chobe National Park in Botswana with Kafue National Park in Zambia. This corridor facilitates the migration and movement of key wildlife species, contributing to biodiversity conservation in the region.

The Simalaha Conservancy is also notable for its community-driven approach to conservation, where local people actively manage natural resources and promote sustainable coexistence between humans and wildlife [10,11]. However, this proximity to wildlife brings challenges, including an increased risk of zoonotic disease transmission, such as anthrax, which thrives in areas of close human-wildlife-livestock interactions. Even though there are many ecological and epidemiological elements that contribute to anthrax epidemics, human-wildlife-livestock interactions are a major study focus because they shape disease dynamics. The ecological and socioeconomic importance of this region makes it an ideal site for studying the complex dynamics of anthrax transmission and identifying interventions that can balance public health and conservation goals.

The other study sites are Lochinvar and Blue Lagoon National Parks, located within the Kafue Flats in Zambia (Fig 1). We specifically selected these study sites due to the endemicity of anthrax in areas surrounding most national parks, highlighting their importance in understanding disease dynamics at the human-wildlife-livestock interface.

Lochinvar National Park, situated on the southern edge of the Kafue Flats, encompasses a vast floodplain formed by the Kafue River between the Itezhi-Tezhi dam to the west and the Kafue Gorge to the east. The park is renowned for its rich biodiversity, particularly its significant populations of Kafue lechwe, a semi-aquatic antelope species endemic to Zambia, and a wide variety of bird species. Its ecological importance does, however, come with drawbacks, such as the overall risk of zoonotic disease transmission in regions where interactions between people, animals, and cattle are common. Environmental features include floodplain ecosystems with seasonal water changes, nutrient-rich soils that allow spore persistence, and carcass decomposition dynamics that facilitate bacterial amplification are particularly important in anthrax epidemics [12].

Blue Lagoon National Park, located approximately 120 kilometres west of Lusaka on the Kafue Flats, shares similar ecological and conservation importance. It serves as a critical habitat for wildlife, including waterbirds and other herbivore species, while also supporting local communities through tourism and ecosystem services [13]. The proximity of both parks to human settlements may increase the likelihood of interactions between wildlife, livestock, and humans, which may amplify the potential for anthrax outbreaks.

### 2.4. Sample size estimation

**2.4.1. Quantitative data.** We lacked precise information on the total number of households in each village. Therefore, we used the sample size estimation formula for unknown populations in Epitools (http://epitools.ausvet.com.au/). Assuming unknown household population in study locations, a confidence level of 95%, estimated proportion of 50%, and desired precision of 5%, the necessary sample size was calculated at 385 respondents, assuming random sampling. Since there were 3 study sites, we multiplied 385 by 3 to get a sample size of 1,155 households.

**2.4.2. Qualitative data.** The sample size for focus group discussions (FGD) was six [6] comprised of 3 women and 3 men's groups. The sample size (N) was arrived at after reaching a data saturation point for focus group discussions. Strauss and Corbin [14], state that Researchers should continue data collection until no new or relevant information emerges within a given category. Data saturation was assessed through an iterative process, involving continuous analysis of data as it emerged from each focus group discussion (FGD). This process included regularly reviewing and interpreting the collected data.

### 2.5. Data collection techniques

**2.5.1. Quantitative data.** We conducted three online meetings to develop a structured questionnaire and plan data collection across three Game Management Areas. The questionnaire was initially drafted in English and circulated among all authors for feedback. This served as a pretest to evaluate its clarity, strengths, and weaknesses, as well as its ability to capture the intended responses. After incorporating the authors' input, we exported the questionnaire into ZOHO software (https://survey.zoho.com/survey) and shared the link with the 26 recruited data collectors (veterinary assistants, agricultural extension officers, and game scouts) through a social media group. This provided an additional layer of pretesting to ensure data quality.

We refined the online questionnaire based on feedback from data collectors, who shared insights on how farmers might interpret the questions. Subsequently, we organized three one-day workshops in each study location to train the data collectors. During these workshops, we observed each data collector administer at least five questionnaires using both online and offline versions on electronic devices.

We piloted the questionnaire in Monze district, which revealed redundancies and areas requiring clarification. Following this pilot, we revised the questionnaire to eliminate repetitive and ambiguous questions. Once finalized, we initiated data collection across all study locations, incorporating input from researchers, farmers, and enumerators. During face-to-face interviews, the questionnaire was translated into the relevant local languages—Tonga, Ila, and Lozi—to ensure effective communication and accurate data collection.

**2.5.2. Qualitative data.** We developed a checklist FGD guide to maintain consistency across discussions and allow flexibility. The guide contained open-ended questions that allowed participants to share their knowledge of anthrax, including symptoms, transmission, and prevention; the influence of cultural beliefs and practices on perceptions of anthrax; the economic and social impacts of anthrax outbreaks on livelihoods and community dynamics; and the perceived role of wildlife as potential sources of anthrax.

We then asked district veterinary officers to identify gate keepers in each community who would recruit members of the focus group meetings and organise venues at respective schools in each study site. Gatekeepers represent the link between the facilitators of focus group meetings and stakeholders and thus play a brokering role and control access to the community [15]. The gatekeepers were community leaders from the traditional leadership who recruited 6–12 women and men for focus group meetings. The selection of participants for the focus group meetings was left to the gatekeepers, with the guidance to include active members of the community. Participants in the focus group meetings were not involved in the cross-sectional survey or discussion by the facilitators in order to avoid them coming with preconceived ideas to the meeting. The agenda of the focus group meetings was shared with gatekeepers who shared it with the participants of the focus group discussions. Each meeting started with scene preparation with a sitting arrangement in a C-shaped form as recommended by similar studies [15–18]. Each FGD was a minimum of 60 minutes to a maximum of 90 minutes. The discussions were in the daily language of each participating community (Silozi, Ila, and Tonga).

## 2.6. Data management and analysis

**2.6.1. Quantitative data.** We downloaded the collected data from ZOHO software in Microsoft Excel format. We also downloaded a full report for results for frequencies which were automatically generated by ZOHO software in pdf format. The data in Microsoft Excel was cleaned and exported to R software version 4.2.1 for analysis. Descriptive statistics, such as frequencies and percentages, were used to summarize socio-demographic characteristics, livelihood, and alternative livelihood. All analyses were conducted using the dplyr R package to ensure accurate computation and visualization

**2.6.2 Qualitative data.** We downloaded audio recordings from the recorder to computers and named the audio files based on the location and gender of the group. Independent native speakers of the local languages (Silozi, Ila, and Tonga) transcribed the recordings verbatim and translated them into English for analysis. One researcher manually coded the data after carefully reading and re-reading the narratives. The initial coding involved broad categorization based on significant themes derived from the topic guides in the checklist. Subsequently, we conducted more detailed coding to break the major themes into sub-themes.

We created framework matrices by extracting key quotes under each sub-theme and summarizing the narratives under each major theme. These matrices facilitated cross-checking of findings from the cross-sectional survey with the narratives from focus group participants, enabling us to identify divergent or supporting perspectives. To maintain anonymity, we replaced identities in the raw narratives with placeholders (e.g., A1) in the results section.

## 3. Results

### 3.1. Results for quantitative data

**3.1.1. Socio-demographic characteristics.** The study revealed that a larger proportion of respondents were male (57.7%), while females accounted for 42.3% as shown in Table 1. The majority of respondents fell within the 36–40 age group, indicating that middle-aged adults were the most represented demographic in the study. Most participants (65.9%) reported being married, reflecting the predominance of family-oriented households in the study population. Regarding educational attainment, 59.9% of respondents indicated having completed primary school, suggesting limited access to higher education among the population.

The findings further indicated that 65.7% of respondents identified as household heads, demonstrating a strong sense of individual responsibility within the sampled communities. Among these household heads, the largest proportion (28.1%)

**Table 1. Socio-demographic characteristics of respondents in study areas.**

| Characteristics | Category | District | | | |
|---|---|---|---|---|---|
| | | **Kazungula** n (%) | **Monze** n (%) | **Mumbwa** n (%) | **Mwandi** n (%) |
| **Age of respondents (N = 1190)** | | | | | |
| | <20 years (n = 56) | 6 (2) | 21 (5.5) | 23 (5.7) | 6 (5.9) |
| | 21-25 (n = 161) | 41 (13.7) | 60 (15.6) | 51 (12.6) | 9 (8.9) |
| | 26-30 (n = 141) | 31 (10.3) | 56 (14.6) | 41 (10.12) | 13 (12.9) |
| | 31-35 (n = 157) | 42 (14) | 40 (10.4) | 59 (14.6) | 16 (15.8) |
| | 36-40 (n = 182) | 40 (13.3) | 66 (17.2) | 62 (15.3) | 14 (13.9) |
| | 41-45 (n = 138) | 32 (10.6) | 47 (12.2) | 40 (9.9) | 19 (18.8) |
| | 46-50 (n = 115) | 31 (10.3) | 32 (8.3) | 46 (11.4) | 6 (5.9) |
| | 51-55 (n = 88) | 22 (7.3) | 23 (6.0) | 30 (7.4) | 13 (12.9) |
| | 56-60 (n = 63) | 18 (6) | 22 (5.7) | 22 (5.4) | 1 (1) |
| | >60 years (n = 89) | 37 (12.3) | 17 (4.4) | 31 (7.7) | 4 (4) |
| **Sex of Respondent (N = 1187)** | Female (n = 502) | 134 (45.0) | 148 (38.5) | 171 (42.2) | 48 (49.5) |
| | Male (n = 685) | 166 (55.0) | 236 (61.5) | 234 (57.8) | 49 (50.5) |
| **Marital Status (N = 1186)** | Divorced (n = 80) | 26 (8.7) | 28 (7.3) | 19 (4.7) | 7 (7.2) |
| | Married (n = 782) | 186 (62.0) | 261 (70.0) | 277 (68.4) | 58 (59.8) |
| | Single (n = 235) | 67 (22.3) | 63 (17.4) | 78 (19.3) | 27 (27.8) |
| | Widowed (n = 89) | 21 (7.0) | 32 (8.3) | 31 (7.6) | 5 (5.2) |
| **Respondent's Status in the household (N = 1184)** | Child of Household Head (n = 97) | 8 (2.7) | 49 (12.8) | 39 (9.6) | 1 (1) |
| | Dependent (n = 46) | 2 (0.7) | 11 (2.9) | 31 (7.7) | 2 (2) |
| | Household Head (n = 778) | 212 (70.7) | 242 (63.4) | 252 (62.2) | 72 (74.2) |
| | Spouse of Household Head (n = 263) | 78 (26.0) | 80 (20.9) | 83 (20.5) | 22 (22.7) |
| **Household Size N = 1186** | 2-3 members (n = 202) | 94 (31.3) | 35 (9.1) | 49 (12.1) | 24 (24.7) |
| | 4-5 members (n = 333) | 105 (35.0) | 87 (22.7) | 106 (26.1) | 35 (36.1) |
| | 6-7 members (n = 300) | 59 (19.7) | 97 (25.2) | 117 (28.8) | 27 (27.8) |
| | 8-9 members (n = 175) | 25 (8.33) | 68 (17.7) | 76 (18.8) | 6 (6.2) |
| | >10 members (n = 176) | 17 (5.6) | 97 (25.2) | 57 (14.1) | 5 (5.2) |
| **Educational Level (1193)** | No formal education (n = 87) | 22 (7.3) | 24 (6.3) | 40 (9.9) | 1 (1.0) |
| | Primary education (n = 711) | 183 (61) | 250 (65.1) | 210 (51.9) | 68 (66.0) |
| | Secondary education (n = 320) | 91 (30.3) | 87 (22.7) | 114 (29.7) | 28 (27.2) |
| | Tertiary education (n = 54 | 4 (1.3) | 15 (3.9) | 34 (8.9) | 1 (1.0) |
| | Vocational (skills) (n = 21) | 0 | 8 (2.1) | 7 (1.8) | 6 (5.8) |

oversaw families with 4–5 members, followed closely by those managing households with 6–7 members (25.3%). These figures highlight the prevalence of moderately sized families in the study area.

**3.1.2. Livelihood and alternative livelihood.** The findings revealed that crop farming was the primary source of income for most respondents, with 42.2% relying on it as their main livelihood as shown in Table 2. Livestock farming emerged as an important alternative income source, reported by 34.7% of participants. Despite these economic activities, the majority of respondents reported monthly incomes below ZMW 1,000 (USD35), highlighting the limited financial resources within the study population.

When asked about household savings, most respondents (61.7%) indicated that they did not have any form of savings. Among the 38.3% who reported having savings, mobile money platforms (41.5%) and village banking systems (40.4%) were the most common methods for saving.

**Table 2. Livelihood and alternative livelihoods.**

| Characteristics | Category | N(%) |
|---|---|---|
| **Main source of income (n-1169)** | Formal employment | 84 (7.19%) |
| | Informal employment | 224 (19.16%) |
| | Crop farming | 493 (42.17%) |
| | Livestock farming | 170 (14.54%) |
| | Bee farming | 0 (0.00%) |
| | Fish farming | 90 (7.70%) |
| | Hunting | 0 (0.00%) |
| | Logging | 6 (0.51%) |
| | Charcoal burning | 86 (7.36%) |
| | Others | 16 (1.37) |
| **Alternative livelihood (n-1197)** | Formal employment | 20 (1.705) |
| | Informal employment | 107 (9.08%) |
| | Crop farming | 412 (34.94%) |
| | Livestock farming | 409 (34.69%) |
| | Bee farming | 8 (0.68%) |
| | Fish farming | 134 (11.37%) |
| | Hunting | 21 (0.76%) |
| | Logging | 9 (1.78%) |
| | Charcoal burning | 288 (24.43%) |
| | Others | 315 (26.72%) |
| **Household income (n-1179)** | Below ZMW 1000 | 624 (52.57%) |
| | ZMW 1100–2000 | 301 (25.36%) |
| | ZMW 2100–3000 | 109 (9.18%) |
| | ZMW 3100–4000 | 47 (3.96%) |
| | ZMW 4100–5000 | 34 (2.86%) |
| | Above ZMW5000 | 72 (6.07%) |
| **Does your household have savings?** | Yes | 455 (38.33%) |
| | No | 732 (61.67%) |
| **How do you keep your savings? (n-1187)** | Village banking | 184 (40.44%) |
| | Commercial Banking | 56 (12.31%) |
| | Mobile banking | 189 (41.54%) |
| | Others | 16 (5.71%) |
| **Is anyone in the family receiving social cash transfer? (n-1187)** | Yes | 350 (29.49%) |
| | No | 837 (70.51%) |
| **Has your household been affected by prolonged drought? (n-1187)** | Yes | 1044 (87.95%) |
| | No | 143 (12.08) |
| **If yes, how do you cope with prolonged dry spell? (n-1153)** | Trading in game meat | 8 (0.485) |
| | Hunting | 18 (1.74) |
| | Livestock farming | 395 (38.69%) |
| | Charcoal burning | 328 (31.69%) |
| | Fishing | 186 (17.97%) |
| | Logging | 6 (0.68%) |
| | Gardening | 417 (40.19) |
| | Others | 242 (23.38%) |

*(Continued)*

**Table 2.** (Continued)

| Characteristics | Category | N(%) |
|---|---|---|
| Support mechanism (n-1185) | Food security pack | 81 (6.84%) |
| | Emergency draught response (COMDEV) | 53 (4.47%) |
| | Relief Food and response (WFP) | 155 (13.08%) |
| | Wetlands seed packs | 13 (1.10%) |
| | None of the above | 819 (69.11%) |
| | others | 96 (8.10) |

A significant proportion of respondents (87.9%) reported experiencing the adverse effects of dry spells in the past, which had a notable impact on their livelihoods. To cope with these challenges, many households turned to alternative activities such as livestock farming (38.7%), charcoal burning (31.7%), fishing (18%), and gardening (40.2%). The types of livestock farmed included, cattle, goats, sheep, chickens, rabbits, and pigs.

The results further indicated that 29.5% of households benefited from social cash transfer programs, suggesting limited access to government-led social protection initiatives. Additionally, respondents reported receiving minimal support from government, NGOs, and donors to cope with drought conditions. Relief food provided by the World Food Programme (WFP) was identified as the primary form of external assistance, but it was only received by 13.1% of respondents as shown in Table 2.

### 3.2. Results for qualitative data: Food security and livelihoods

This theme examines the interconnectedness of food security and livelihoods within the context of anthrax. It explores the diverse food sources relied upon by communities, the impact of anthrax on food availability and access, and the potential for diversifying livelihoods to enhance resilience.

3.2.1. **Food sources.** The data reveals a reliance on diverse food sources, including livestock, crops, and wild foods. Game meat, in particular, plays a significant role in the diets and livelihoods of many communities, especially those living in close proximity to wildlife areas.

*Kazungula Men*: A2: *"We depend on fishing and farming. We also hunt for food, especially during the dry season when crops are scarce"*.

*Mumbwa Women*: A3: *"We mostly rely on crop farming for our food. But we also keep some chickens and goats for meat and milk"*.

The diversity of food sources reflects the adaptive strategies employed by communities to ensure food security in different seasons and environmental conditions. However, the reliance on game meat also highlights the potential vulnerability to anthrax outbreaks, which can disrupt access to this important food source.

3.2.2. **Impact of anthrax on food security.** Anthrax outbreaks can have a significant impact on food security, particularly when they result in livestock deaths and restrictions on meat consumption. This can lead to reduced protein intake, nutritional deficiencies, and increased vulnerability to hunger and poverty.

*Kazungula Women*: A4: *"When the anthrax outbreak happened, we lost many of our cattle. This made it very difficult for us to feed our families. We had to rely on government assistance for food"*.

*Mumbwa Men*: A5: *"The restrictions on meat sales during the anthrax outbreak affected our income. We could not sell our livestock and we had to spend money on buying food that we normally get from our animals".*

The impact of anthrax on food security underscores the need for interventions that not only control the disease but also support communities in coping with its consequences. This may involve providing food aid, promoting alternative food sources, and strengthening social safety nets.

**3.2.3. Alternative livelihoods.** Participants highlighted the importance of diversifying livelihoods to reduce dependence on livestock and game meat, thereby enhancing resilience in the face of anthrax outbreaks and other shocks. They expressed interest in exploring alternative income-generating activities, such as beekeeping, poultry farming, and small-scale businesses.

*Kazungula Men*: A6: *"We need to find other ways to make a living so that we are not solely dependent on our livestock. If we have other sources of income, we can better cope with challenges like anthrax outbreaks".*

*Mumbwa Women*: A7: *"I would like to learn new skills so that I can start a small business. This would help me to earn money and provide for my family, even if we lose our livestock to anthrax".*

Promoting diversification of livelihoods can enhance community resilience and reduce vulnerability to food insecurity. This requires providing training, access to credit, and market opportunities for alternative income-generating activities [19].

## 3.3. Cross-checking of findings from the cross-sectional survey with the narratives from focus group participants

**3.3.1. Socio-demographic characteristics and livelihood context.** The quantitative findings revealed that the majority of respondents were middle-aged, married, and primarily engaged in crop farming, with livestock farming serving as an alternative livelihood source. Additionally, limited access to higher education and formal savings mechanisms characterized the study population.

Qualitative data provided depth to this understanding, as participants emphasized their dependence on subsistence farming, livestock, and informal income sources. For instance, one participant stated:

*"We depend on fishing and farming. We also hunt for food, especially during the dry season when crops are scarce."*

**3.3.2. Food security and the impact of anthrax.** Quantitative findings indicated that 87.9% of respondents experienced the adverse effects of drought, which exacerbated food insecurity. Households relied heavily on adaptive strategies such as gardening (40.2%), charcoal burning (31.7%), and fishing (18%). Furthermore, the quantitative data revealed that 69.1% of respondents did not receive external drought-related assistance, underscoring the community's vulnerability.

The qualitative data added nuance, illustrating how anthrax outbreaks compounded food insecurity. For instance, one respondent shared:

*"When the anthrax outbreak happened, we lost many of our cattle. This made it very difficult for us to feed our families. We had to rely on government assistance for food."*

This integration underscores the dual challenges of climatic shocks and zoonotic diseases on food security. It demonstrates the need for comprehensive interventions to address both immediate and systemic drivers of food insecurity.

**3.3.3. Livelihood diversification as a coping strategy.** The quantitative data revealed that households employed a range of coping mechanisms, including charcoal burning (31.7%) and livestock farming (38.7%). However, less than 30% of households benefited from social cash transfer programs or other forms of support.

Qualitative insights reinforced the need for livelihood diversification, as participants expressed interest in beekeeping, poultry farming, and small-scale businesses to reduce reliance on livestock and game meat:

*"We need to find other ways to make a living so that we are not solely dependent on our livestock."*

The integration of data highlights the importance of empowering communities with skills, resources, and market access to pursue alternative livelihoods. This may also reduce poaching wildlife for survival among communities living in Game Management Areas

## 4. Discussion

The study explored the relationship between alternative livelihoods in Game Management Areas (GMAs) and livestock farmers' exposure to anthrax, revealing critical insights into the socio-economic and ecological dynamics at play. The demographic profile of respondents indicates that the majority were male (57.2%), a pattern often linked to men's role in livestock herding and management [20]. Additionally, 65.9% of respondents were married, which may reflect the prioritization of livestock as a family asset. Married farmers, driven by family dependencies, might engage in high-risk livestock management practices despite anthrax-related dangers [21]. For instance, farmers who practice the ''mafisa'' system, which involves loaning cattle to others for herding, like in the Western Province of Zambia, often lead animals being moved across numerous areas, increasing the potential for disease spreading, including anthrax, while cattle move through contaminated areas [22].

Education levels varied among respondents, with most having some form of formal education. Educated farmers are more likely to seek veterinary advice and vaccinate their animals, which helps mitigate disease risks [23]. However, 7.33% of respondents had no formal education, which may limit access to critical information and veterinary services, increasing the likelihood of anthrax exposure.

Family sizes were predominantly below six members, with an average slightly lower than the national average of six [24]. Larger households may have more members engaged in livelihood activities, increasing exposure to anthrax-prone environments. Livelihood diversification among farmers in GMAs includes crop farming, charcoal burning, fishing, livestock rearing, and small businesses. While diversification offers financial stability, activities such as charcoal burning disturb anthrax spores in soil, increasing exposure risks [25]. Charcoal burning aids in exposing the anthrax spores as it involves, cutting of trees, uprooting of roots and disturbing top soils which can expose dormant anthrax and make them more accessible to human and livestock and can alter soils condition [26]. Furthermore, burning charcoal causes deforestation, which diminishes natural habitats and brings wildlife closer to areas utilized for cattle grazing and human habitation. Increased contact may make it easier for anthrax to spread, particularly if the bacteria is present in wildlife or livestock. According to a research conducted in Chama District, in the Eastern Province of Zambia, where an anthrax outbreak occurred and resulted in 511 human cases and 85 hippopotamus deaths, environmental changes and food deprivation caused animal and human activity to come into closer contact [27]. A significant proportion of respondents reported household savings below ZMW1,000 (USD 35), primarily managed through mobile and village banking. According to the Zambia Statistics Agency's living conditions monitoring survey, the average monthly household income in rural areas was estimated to be K2112.20 (less than $100), while the poverty threshold revealed that 60% of Zambians lived below the national poverty line, with a range of 78.7% in rural areas and 31.9% in urban areas [28]. These savings are primarily managed through mobile and village banking. Limited financial buffers exacerbate vulnerability to zoonotic diseases like anthrax, as poverty constrains investments in preventive measures such as vaccination of livestock against diseases like anthrax [29]. Much as income diversification enhances resilience, it can also expose farmers to new risks, including anthrax, due to increased interactions with wildlife in GMAs such as butchering of animals (wild or domestic) that have died of anthrax, or digging in the ground and creating spore-containing aerosols, would be potential ways of contracting anthrax in this setting [22,30].

Farmers demonstrated varying levels of awareness regarding anthrax. Informal networks, including family and local veterinary officers, were the primary sources of information. While community-based information dissemination can be effective, gaps in knowledge regarding preventive measures and vaccination protocols remain. Previous studies highlight that awareness alone is insufficient, as economic pressures often compel farmers to engage in risky practices, such as handling carcasses of animals that died from unknown causes [31]. Targeted awareness campaigns, coupled with livelihood support, could significantly reduce anthrax risks [32].

The study revealed that 87.95% of farmers experienced prolonged dry spells, increasing competition for scarce resources and driving interactions between livestock and wildlife. Such conditions often force farmers to adopt coping strategies, including gardening, hunting, fishing, and trading game meat. However, these activities heighten anthrax exposure risks. For instance, gardening may disturb dormant anthrax spores in contaminated soil, while hunting and fishing increase contact with wildlife reservoirs of anthrax [26,31]. Although gardening typically requires access to water, participants identified it as a viable coping strategy during dry spells due to the availability of residual moisture near riverbanks, floodplains, or hand-dug wells in certain locations. In these areas, communities were able to grow fast-maturing vegetables or drought-tolerant crops on a small scale for household consumption and local sale. This form of gardening served as a supplementary food and income source, particularly when rainfall-dependent crops failed. However, its feasibility varied depending on local water access and was not uniformly available across all sites.

Alarmingly, 69.11% of respondents reported having no formal support mechanisms during drought, while only a few relied on government support, such as food security programs and emergency drought responses. Resource scarcity during climatic shocks has been shown to increase reliance on high-risk practices, further heightening anthrax exposure. While managing livestock in areas where anthrax is widespread is inherently risky, households may unintentionally become more vulnerable to the disease as a result of their increasing reliance on alternative sources of income like fishing, charcoal burning, and small-scale enterprises. For instance, in the recent anthrax outbreaks in Southern Province, fishermen were the first ones to be involved in butchering floating hippos that died of anthrax (Personal communication). This is because the risk of outbreaks may increase if focus, effort, and resources are taken away from vital animal care procedures, including vaccination, disease surveillance, and pasture management. Studies emphasize the need for social safety nets to reduce dependence on high-risk practices during crises [33]. Additionally, a study conducted in the Western Province of Zambia revealed that anthrax outbreaks were more likely to occur during dry months when human occupancy of the floodplain increased, suggesting that seasonal migration and reliance on the floodplains for livelihood during dry spells contribute to heightened anthrax risks [34].

The reliance on immediate income-generating activities such as charcoal burning, fishing, and small-scale businesses often diverts attention and resources away from vaccinating livestock against diseases, increasing the risk of anthrax transmission and maintenance. Studies emphasize the need for social safety nets to reduce dependence on high-risk practices during crises [33]. However, critics argue that such support systems are often insufficient, leaving gaps that perpetuate vulnerability [26].

## 4.1. Implications of the study on policy

This study has important implications for public health policy, livestock management, and rural development in Game Management Areas in Zambia. The findings demonstrate that livelihood diversification, though essential for economic resilience, may inadvertently increase community exposure to anthrax through high-risk coping strategies such as charcoal burning, fishing, and reliance on contaminated game meat. This underscores the need for integrated and context-specific interventions that align environmental conservation goals with public health and socio-economic stability.

First, the study highlights the critical importance of strengthening disease surveillance and veterinary services in high-risk areas. Improved access to livestock vaccinations and timely outbreak response mechanisms can significantly reduce the burden of anthrax among rural communities. Public health education that links disease prevention with safe

livelihood practices is equally essential, particularly in areas where informal knowledge networks are the primary sources of information.

Second, the data underscore the value of investing in sustainable livelihood alternatives, such as beekeeping, poultry farming, and small-scale enterprises. These alternatives offer pathways to reduce dependence on risky practices while improving household income and food security. However, successful implementation requires enabling conditions, including access to training, microfinance, markets, and infrastructure.

Third, the gendered dimensions of vulnerability revealed by this study call for women-centered programming. Women often bear the brunt of food insecurity and lack access to resources for livelihood diversification. Tailored support for women's economic empowerment and participation in disease prevention programs is essential to strengthen household and community resilience.

Finally, the study illustrates the value of mixed-methods approaches in understanding complex health-environment-livelihood systems. Integrating quantitative data with qualitative insights provided a richer understanding of the socio-cultural and economic drivers behind anthrax exposure, enabling more targeted and actionable recommendations.

### 4.2. Study limitations and future research

While this study provides valuable insights into the socio-economic and ecological dynamics influencing anthrax exposure in Zambia's GMAs, several limitations must be acknowledged. First, the reliance on self-reported data through structured questionnaires and focus group discussions introduces potential recall and reporting biases.

Second, the study was conducted in selected sites within Southern and Western Provinces, which may limit the generalizability of the findings to other GMAs or regions with different ecological or socio-cultural contexts.

Third, while the quantitative sample size was statistically estimated, qualitative insights were derived from a limited number of focus groups, which may not fully capture the diversity of experiences across communities.

Lastly, logistical and resource constraints restricted deeper investigation into environmental sampling or pathogen surveillance, which would have strengthened the ecological inferences. Future research should explore this aspect.

### 4.3. Conclusion and recommendations

This study demonstrates the critical interconnectedness of anthrax, food insecurity, and livelihood strategies in rural Zambian communities living within Game Management Areas. While alternative livelihoods such as charcoal burning, fishing, and informal trade provide economic buffers during climatic and disease shocks, they also inadvertently increase exposure to anthrax through heightened contact with contaminated environments and wildlife. Tailored interventions emphasizing community empowerment, gender-sensitive livelihood diversification, and improved access to veterinary and extension services are urgently needed to mitigate zoonotic disease risks and strengthen resilience. A One Health approach that integrates public health, animal health, and environmental sustainability is essential for reducing vulnerabilities in these high-risk settings.

### Acknowledgments

The authors are grateful to staff members of the Department of Veterinary Services in the Ministry of Livestock and Fisheries in Monze, Kazungula and Mumba District for facilitating data collection. AI was used for qualitative data analysis and to improve the language of this article: OpenAI. (2024). ChatGPT [Large language model]. https://chatgpt.com.

### Author contributions

**Conceptualization:** Laila Gondwe, Chisoni Mumba, Exillia Kabbudula, Suwilanji S. Sichone, Mwila Kayula, Geoffrey Mainda, Fredrick M. Kivaria, Charles Bebay, Baba Soumare, Mtui-Malamsha J. Niwael, Chitwambi Makungu.

**Data curation:** Laila Gondwe, Chisoni Mumba, Kezzy Besa, Exillia Kabbudula, Noanga Mebelo.

**Formal analysis:** Laila Gondwe, Chisoni Mumba, Kezzy Besa, Davies Phiri, Exillia Kabbudula, Noanga Mebelo.

**Funding acquisition:** Chitwambi Makungu.

**Investigation:** Laila Gondwe, Chisoni Mumba, Exillia Kabbudula, Suwilanji S. Sichone, Chitwambi Makungu.

**Methodology:** Laila Gondwe, Chisoni Mumba, Exillia Kabbudula, Chitwambi Makungu.

**Project administration:** Chisoni Mumba, Chitwambi Makungu.

**Resources:** Chisoni Mumba, Mtui-Malamsha J. Niwael.

**Software:** Laila Gondwe, Chisoni Mumba, Davies Phiri.

**Supervision:** Chisoni Mumba, Chitwambi Makungu.

**Validation:** Laila Gondwe, Chisoni Mumba, Kezzy Besa, Davies Phiri, Exillia Kabbudula, Noanga Mebelo, Suwilanji S. Sichone, Ntazana N. Sinyangwe, Mwila Kayula, Geoffrey Mainda, Fredrick M. Kivaria, Charles Bebay, Baba Soumare, Mtui-Malamsha J. Niwael, Chitwambi Makungu.

**Visualization:** Laila Gondwe, Chisoni Mumba, Kezzy Besa, Davies Phiri, Exillia Kabbudula, Noanga Mebelo, Suwilanji S. Sichone, Ntazana N. Sinyangwe, Mwila Kayula, Geoffrey Mainda, Fredrick M. Kivaria, Charles Bebay, Baba Soumare, Mtui-Malamsha J. Niwael, Chitwambi Makungu.

**Writing – original draft:** Laila Gondwe, Chisoni Mumba, Kezzy Besa, Davies Phiri, Exillia Kabbudula, Noanga Mebelo, Suwilanji S. Sichone, Ntazana N. Sinyangwe, Mwila Kayula, Geoffrey Mainda, Fredrick M. Kivaria, Charles Bebay, Baba Soumare, Mtui-Malamsha J. Niwael, Chitwambi Makungu.

**Writing – review & editing:** Laila Gondwe, Chisoni Mumba, Kezzy Besa, Davies Phiri, Exillia Kabbudula, Noanga Mebelo, Suwilanji S. Sichone, Ntazana N. Sinyangwe, Mwila Kayula, Geoffrey Mainda, Fredrick M. Kivaria, Charles Bebay, Baba Soumare, Mtui-Malamsha J. Niwael, Chitwambi Makungu.

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
