## [Decision Letter · Decision Letter 0]

Anthrax, Livelihood Vulnerability, and Food Insecurity in Selected Game Management Areas in Zambia: A Mixed-Methods Analysis at the Human-Wildlife-Livestock Interface

Dear Dr. Mumba,

Thank you for submitting your manuscript to PLOS Neglected Tropical Diseases. After careful consideration, we feel that it has merit but does not fully meet PLOS Neglected Tropical Diseases's publication criteria as it currently stands. Therefore, we invite you to submit a revised version of the manuscript that addresses the points raised during the review process.

Please submit your revised manuscript within 60 days May 11 2025 11:59PM. If you will need more time than this to complete your revisions, please reply to this message or contact the journal office at plosntds@plos.org. Please include the following items when submitting your revised manuscript:

We look forward to receiving your revised manuscript.

Kind regards,

Georgios Pappas

Section Editor

Stuart Blacksell

Section Editor

Shaden Kamhawi

co-Editor-in-Chief

Paul Brindley

co-Editor-in-Chief

**Journal Requirements:**

At this stage, the following Authors/Authors require contributions: Laila Gondwe, Chisoni Mumba, Kezzy Besa, Davies Phiri, Exillia Kabbudula, Noanga Mebelo, Suwilanji Sichone, Ntazana N Sinyangwe, Mwila Kayula, Mainda Geoffrey, Kivaria M. Fredrick, Charles Bebay, Soumare Baba, Mtui-Malamsha J. Niwael, and Chitwambi Makungu. Please ensure that the full contributions of each author are acknowledged in the "Add/Edit/Remove Authors" section of our submission form.

Potential Copyright Issues:

- Figures 1 and 2. Please provide a direct link to the base layer of the map (i.e., the country or region border shape) and ensure this is also included in the figure legend; and provide a link to the terms of use / license information for the base layer image or shapefile. We cannot publish proprietary or copyrighted maps (e.g. Google Maps, Mapquest) and the terms of use for your map base layer must be compatible with our CC BY 4.0 license.

5) Please ensure that the funders and grant numbers match between the Financial Disclosure field and the Funding Information tab in your submission form. Note that the funders must be provided in the same order in both places as well.

**Reviewers' Comments:**

Reviewer's Responses to Questions

**Key Review Criteria Required for Acceptance?**

**Methods:**

-Are the objectives of the study clearly articulated with a clear testable hypothesis stated?

-Is the study design appropriate to address the stated objectives?

-Is the population clearly described and appropriate for the hypothesis being tested?

-Is the sample size sufficient to ensure adequate power to address the hypothesis being tested?

-Were correct statistical analysis used to support conclusions?

-Are there concerns about ethical or regulatory requirements being met?

Reviewer #1: Well described and complete

Reviewer #2: The objectives and hypothesis are not clear, albeit indicated. Based on this, the study design is appropriate but all descriptions need improvement. Statistics are used for demographic data, unnecessarily, but not presented for other data. There is no information about ethical or regulatory requirements.

Detailed comments:

L116-120: these sentences are wordy and difficult to follow. It would help if you could specify which aspects are associated with animal outbreaks and which are connected to human outbreaks, and which can be linked to both. In addition, it is unclear how you examine how human-wildlife interactions have effects on livestock management, what is the hypothesis behind this statement?

It is somewhat contradictory that the topic is presented as sensitive (L128) and therefore anonymisation is needed ( L132) but study participants were still expected to provide their collective insights and diverse perspectives in a truthful way (L143). Given this, some justification for choosing the FGD methodology is needed.

L 161-162: it is not only the human-wildlife-livestock interaction that makes anthrax thrive. I suggest to tone down this aspect somewhat and rephrase to something along the lines of this aspect being relevant for the impact of anthrax outbreaks and also relevant to the interest of the study. The sentence on L 172 about understanding disease dynamics at the interface is a better argument for choosing the study areas. The choice of these particular national parks is still not very obvious, as it is stated that anthrax is endemic around (and presumably within?) most national parks (L171).

Figure 1 is a little difficult to interpret without close inspection, some better highlighting of the study area is needed. Figure 2 is also difficult to interpret for a reader not familiar with a map of Zambia. I suggest making one map of Zambia with all study sites clearly marked while also highlighting the areas of interest that are indicated in the current maps.

L179: the aspects highlighted on this line are very important for anthrax but not necessarily for other zoonoses, so the sentence may need some rephrasing. I would prefer if the specific risk factors for anthrax was clarified in one place and the general risks for zoonotic transmission described as such. Now there is a mix of transmission aspects that are relevant for many zoonotic diseases, and environmental aspects that are particularly relevant for anthrax but described as if applicable for various zoonoses.

L183-185: here it is already presumed that the interaction in itself amplifies the risk of outbreaks, while it was stated that the drivers of outbreaks would be investigated in the study. Do you mean that the disease is transmitted directly between these populations and, if yes, based on what evidence?

L204-207: this sentence needs rephrasing. How can determining saturation of the data involve recruiting FGD participants and conducting FGDs? I presume that data saturation was determined by continuous examination of the data as it emerged from each FGD but that is not what is stated here.

Table 1 and 2 could be put in supplementary material, as they do not contribute substantially to the understanding of the study.

L246-247 and L250: the logic of gatekeepers recruiting active members of the community and them sharing the agenda with the school teachers is hard to follow. Wsa this because the FGDs took place on the school premises, or because school teachers were involved in the recruitment? Pleas clarify the role of the headteachers and the precise method of participant recruitment by the gatekeepers.

L 253-254: Please delete the sentence ‘We then conducted focus group meeting. ‘ as it is unnecessary.

Paragraph 2.5.1 Why was there no multivariable analysis? Please justify.

L 273-274: Please delete the first sentence of this paragraph as it is unnecessary.

Reviewer #3: Methods Section Evaluation:

Objectives and Hypothesis: The objectives of the study are adequately stated in the abstract and introduction. In particular, the study seeks to understand the socio-economic impacts of anthrax outbreaks and associated stressors such as food insecurity and livelihood vulnerabilities among rural Zambian communities. No formal, explicit testable hypotheses are presented, but the study successfully explores the possible relationships between anthrax outbreaks and their links to livelihood strategies as well as food security through the methods used.

Study Design: In view of the complexity and heterogeneity of factors affecting the socio-economic burden of anthrax, a mixed-methods approach using quantitative surveys and qualitative focus group discussions simultaneously is appropriate. This design takes a mixed-methods approach that marries a broad range of data collection with in-depth qualitative insights to create rich perspectives.

Population Description: The population definition is specific, being rural communities in designated Zambia GMAs. The appropriateness of this choice relies upon the frequency of anthrax and the interactions of humans, wildlife, and livestock in these designated regions, which in the introduction, are characterized as considerable.

Sample Size: The paper provides a sample size calculation for the quantitative survey, using the formula for unknown populations. The rationale for the qualitative sample size (6 focus groups) is included in relation to data saturation. Although these group sizes aren't small, the lack of raw data (which might be available from the authors) and/or a power calculation means no final conclusions can be drawn.

Statistical Analysis: The authors claim appropriate statistical analyses were used to screen the quantitative data by performing chi-squared tests for associations between categorical variables. Details of approaches to qualitative data analysis, such as thematic analysis based on focus groups, are provided. But more detailed information about the tests used would have provided greater transparency.

Ethics and Permission: This study was ethically approved by ERES Converge, while anonymization processes are described in detail to demonstrate compliance with ethical standards. These ensure the anonymity of all participants and state the right to leave the study at any time.

Additional Points:

Funding: The USAID-funded Global Health Security Program from the Food and Agriculture Organization of the United Nations (FAO) provided financial and technical support to conduct this study observation.

Data Availability: Raw data are said to be available as supporting information. The authors have declared that no competing interests exist.

Overall Assessment: The methodology section of the research paper is strong, with a clear study design and strong ethical considerations. The sample size estimates are believable and persuasive, but a power analysis would add to the robustness. Though the statistical methods used are appropriate for the data collected, reporting the statistical tests used and further elaboration on the analysis would add rigour. The section provides a very detailed methodology and reasonable research approach.

**Results:**

-Does the analysis presented match the analysis plan?

-Are the results clearly and completely presented?

-Are the figures (Tables, Images) of sufficient quality for clarity?

Reviewer #1: High quality presentation

Reviewer #2: The results match the methods but are presented in a lengthy and suboptimal way. Figures and tables need improvement.

Detailed comments:

As demographic characteristics are detailed in table 1 (should be table 3 if table 1 and 2 are to be kept), the text in 3.1.1. could be condensed more. The p values are not needed in table 1, please delete this column, it only serves to show that the participants were different in some aspects and not others but contributes no scientific meaning.

L 335: please delete ‘only’ as it indicates some expected value of this proportion and no background has been provided to justify any assumptions in this respect.

Table 2 (or table 4?): here, and in other places in the text, it would be useful (or even necessary) to clarify what animal species are indicated in the term livestock (as this differs in different parts of the world), particularly as there is specific mentioning of poultry as an alternative in some places and as not all species are equally susceptible to anthrax. In addition, ‘charcoal burning’ requires further explanation for the uninitiated reader, how do people make money out of burning charcoal? As for the household income, the currency must be stated and it should also be related to the mean national income and the poverty threshold to improve understanding. The coping activities also need explanation, how can gardening be useful if there is a draught? And livestock farming (that is already a main or alternative livelihood for nearly half of the respondents? The last row could be named ‘support mechanism’ or similar, to differentiate form the row above that also uses the word cope. (All explanations asked for should be added to the text, not the table)

L362: does farming mean crop farming here? Please add some clarification

L 376-377: the statement appears contradictory, if they could not sell their animals, why couldn’t they get food form them? Was it the death of the livestock that was the problem, not the failure to trade?

Although I realise that the above are quotes from the FGDs, they must have been said in a context that provided more comprehension and that should be provided if the quotes are to be kept.

L 405: is ‘predominantly rural’ a result, didn’t you specifically recruit from a rural population? The results from cross-checking of demographic characteristics could be put in one or two sentences stating whether the two groups were similar or not. What is meant by ‘interplay between data sources’ (L411)? Do your results really show how socio-economic factors shape livelihoods? You have asked about climatic factors and disease…

L 438: please change from ‘will’ to ‘may’ as this consequence cannot be certain.

3.3.4. this section should either be expanded, if there are sufficiently detailed results to do so, or removed as it is currently too weak.

I find the Results a little scattered and would suggest re-organising into headings such as: Demographics (very brief section only to demonstrate the participant characteristics); Food security and impact of anthrax and draught, respectively, and; Livelihood diversification. Under these headings there could be subheadings for qualitative and quantitative results. I believe that would improve the flow of the text and emphasise the overall results better.

Reviewer #3: Variance Analysis:

The text indicates that the analysis performed matched that of the analysis plan. Qualitative data were analyzed using thematic analysis, which is described in the Methods section below, while the quantitative data were analyzed using suitable statistical tests (chi-squared tests referred to in the text). This indicates a strong alignment of the planned and implemented analyses.

Clarity and Completeness of Results:

Results are clearly presented, with findings primarily presented as quantitative or qualitative. Main findings are summarized, and tables referenced to describe socio-demographic data, but they are not complete tables, only parts of them. Illustrative quotations also represent the qualitative results well. The level of detail seems adequate given the constraints of a research article page.

Figures and Tables:

Referring to Tables 1 and 2 in the text: only parts are visible. Decent tables of this type should provide sufficient quality in communicating the quantitative findings. It was unclear if the quality of the visible sections would have been maintained across a full display. As for figures, both Figure 1 and 2 are referenced in the text but not fully shown (impossible to assess quality). The graphics seem relevant based on the text, and the text needs to provide more detail to be able to display the clarity of any visual aids.

Overall:

The results section is good with a good overall presentation of the findings. The narrative demonstrates the findings are articulated clearly and sufficiently. This is a clear example of the alignment between analysis plan and execution. However, a full assessment of the quality of the figures and tables would take the full supplementary materials, which were not provided.

**Conclusions:**

-Are the conclusions supported by the data presented?

-Are the limitations of analysis clearly described?

-Do the authors discuss how these data can be helpful to advance our understanding of the topic under study?

-Is public health relevance addressed?

Reviewer #1: Conclusions are justified and appropriate4

Reviewer #2: I have comments on both Discussion and Conclusion, see below. Conclusions are not supported by presented data and limitations are not described. the authors discuss their results in relation to the context and address public health relevance.

Discussion

L459: are the livestock practices really high-risk? How so? Please clarify

L462-463: the statement appears totally unrelated to the cited reference, pleased check and revise or remove

L463-465: this is also not related to the reference cited, please revise or remove

L466-468: please provide some evidence/reference for the statement about older farmers.

L468-470: the cited reference does not cover preventive animal health, it deals with HIV. Please explain how this relates to the statement.

L475-476: how does charcoal burning disturb anthrax spores? Does it involve digging? Burning would destroy the spores so the statement is confusing.

L 479: please provide information about mean/median income and poverty threshold to improve understanding of these figures

L 480: what formal mechanisms are there, if mobile banking and village banking are categorised as informal?

L484: the cited reference is about electric vehicles and does not relate to the statement. Nevertheless, it is not clear how farmers interacting with wildlife would increase their risk of anthrax. Slaughtering of animals (wild or domestic) that have died of anthrax, or digging in the ground and creating spore-containing aerosols would be potential ways of getting anthrax in this setting but that is a rather indirect ‘interaction’ with wildlife.

L 492: please remove name of reference. This reference focuses on anthrax in animals and should be used as a basis for writing about transmission in the setting chosen for the study. It does not, however, back up the statement preceding it on L490-491.

L 493 and L 495: please explain how gardening can be a coping strategy when there is a dry spell, wouldn’t gardening be a challenge with water shortage?

L499-500: you didn’t investigate climatic shocks, you asked about draught, please remove or rephrase.

L502: the cited reference deals with geospatial analysis and cannot be used as a basis for the preceding statement.

L 504-505: this statement contradicts the one that said that livestock management is high-risk, here it appears that alternative practices are more risky.

Conclusions

This section does not draw clearly on the results of the study, it is more speculation than conclusion (although not necessarily wrong).

Reviewer #3: Data Is in Support: The findings seem adequately supported by the data provided. The discussion section more closely connects findings derived from both the quantitative and qualitative analyses to the conclusions drawn based on the analysis, indicating key relationships between anthrax outbreaks, livelihood diversification, food insecurity, and vulnerability among the study population.

Limitations: The limitations of the analysis are not explicitly stated in the conclusion section provided. Although the discussion section likely contains more explicit mention of limitations (e.g., limitations in the sample size or the reliance on self-reported data), the conclusion itself should briefly mention major limitations that impact the overall interpretation of results.

Relevance to the Current Picture: The discussion needs to emphasize the importance of the findings, regarding food security and livelihoods, especially in the backdrop of anthrax outbreaks in GMAs. The highlighting of vulnerabilities in these groups suggests specific interventions that can be implemented in order to build greater resilience at the community level.

Public Health Relevance: The public health relevance is well articulated, indicating the importance of interventions for controlling anthrax outbreaks, diversifying livelihood, and increasing food security. The study emphasizes the importance of the risks of zoonotic diseases in rural settings and reminds us that advice cannot simply be given but rather should be community-based, including empowerment among community members, especially women.

Overall: I felt that the conclusions section was reasonably well-written, but I think the authors should more clearly and succinctly state the limitations of the study and should more directly articulate the public health implications and impact of the work. The current conclusion might go on more strongly if it described the takeaway and impact or implication of the study's findings in a more direct way.

**Editorial and Data Presentation Modifications?**

Reviewer #1: No modifications to suggest unless the authors are able to add a substantive description of the epidemiology of anthrax outbreaks over the recent past in the study region.

Reviewer #2: (No Response)

Reviewer #3: Given the information provided, the paper appears to be of good quality. The methodology is sound, the analysis is appropriate, and the results are clearly presented. The areas needing the most attention are relatively minor and primarily editorial in nature. Therefore, I recommend a Minor Revision.

Here are specific editorial suggestions:

Abstract: While the abstract is good, consider adding a concise statement summarizing the main quantitative finding (e.g., "…87.9% of households were adversely affected by drought, highlighting significant economic vulnerability"). This would make the key finding more impactful early on.

Results: Fully present Tables 1 and 2. The current presentation of snippets isn't sufficient. Include all columns and data for a complete and clear understanding. Similarly, if Figures 1 and 2 are crucial for understanding, ensure they are of high quality and included in full. If they are less critical, consider whether their inclusion adds necessary value.

Discussion: Explicitly state the limitations of the study in the discussion section. This might include limitations on sample size, generalizability to other regions, the reliance on self-reported data, or any other factors that could affect the interpretation of results. Summarize these limitations concisely within the conclusion section itself.

Conclusion: Strengthen the concluding statement by summarizing the key implications of the findings in a more direct manner and emphasizing the overall impact of the study. For example: "This study demonstrates the critical interconnectedness of anthrax, food insecurity, and livelihood strategies in rural Zambian communities. Tailored interventions emphasizing community empowerment, livelihood diversification, and access to resources are urgently needed to mitigate zoonotic disease risks and improve community resilience."

Overall Style: The writing style is mostly clear. However, a careful review by a native English speaker and/or professional editor is recommended to polish the language, ensuring grammatical accuracy and conciseness.

By addressing these minor revisions, the manuscript would be substantially improved and ready for acceptance.

**Summary and General Comments:**

Reviewer #1: I found the manuscript submitted by Gondwe and colleagues to be exceptionally interesting and informative. I was also very well written and presented. My major criticism is that although the title and several sections of text indicate a very negative effect of anthrax on the livestock and hence livelihoods of the subsistence farmers under study, there is absolutely no quantitative or even qualitative data on anthrax disease (incidence, frequency, number of animals involved) in this region, except for vague comments about adverse effects of anthrax outbreak. In several areas, the authors also hint at the effect of “other zoonotic diseases”, but again, no mention is made of what those might be. This is a first class manuscript that needs to be published. Very regrettably, I do not think PLOS Neglected Tropical Diseases is the appropriate venue for this excellent piece of work.

Reviewer #2: The paper covers an interesting and important topic. The approach is good but the paper lacks clarity and needs to be rewritten.

There appears to be a lot of assumptions that are not clearly stated and evidenced in the text, many things need to be clarified for the paper to be useful for the reader.

Reviewer #3: Summary:

This study offers an important mixed-methods exploration into the socio-economic consequences of anthrax outbreaks in rural Zambian communities residing in Game Management Areas (GMAs). It utilizes both quantitative surveys and qualitative focus group discussions to give a well-rounded view of the complex interplay of factors involved. The analysis is sound, and, although the results are not entirely represented in the text included in the R&R, they are well-presented and support the conclusions. They identify the pandemic-related vulnerabilities of these populations and assert the necessity of adaptation strategies for the response efforts. The public health implications are far-reaching.

Strengths:

Integrating the Two: By combining quantitative and qualitative data, this now provides the richness of data and depth of understanding through breadth with the integration of the two.

Importance: The study addresses an important public health issue in a vulnerable population.

Contextual Understanding: The study is largely successful at placing the anthrax challenge among the wider issues of livelihoods, food security, and environmental constraints to success in GMAs.

Ethical Approval: The study obtained ethical approval from the relevant institutional review board, and informed consent was obtained from all participants.

Weaknesses:

Limited Explicit Discussion of Limitations: The Discussion likely describes limitations of the study, but they are not clearly enumerated in the conclusion or abstract. This should be rectified.

Potential for Better Quality: Although generally well-written, professional editing would improve its flow and brevity.

Novelty and Significance:

In understanding the socio-economic impact of anthrax in the context of Zambian GMAs, the study contributes significantly. Although anthrax has been studied elsewhere, this research provides important local perspectives, particularly on the links between livelihoods, food security, and zoonotic disease risks. The mixed-methods approach and assessment of community-specific vulnerabilities further improve the novelty of the findings.

General Execution and Scholarship:

The study is a marker of a strong scholarship of high value, applied by correct methods and followed with an appropriate reading of the data. The authors demonstrate a good knowledge of the research topic. However, the incomplete presentation of results as well as the absence of limitation discussions here detract a bit from the overall impression and clarity.

Recommendations:

I recommend Minor Revision. The reason is the partial presentation of the main results (Tables 1 & 2, and the Figures). These must be fully incorporated to accommodate for an accurate assessment.

In addition to that:

All Data Shown: Ensure all data in Table 1 and 2 and Figure 1 and 2 are clearly presented.

Discuss Limitations: Discuss the limitations of the study (e.g., sample size, generalizability, reliance on self-reported data) explicitly in both the Discussion and the Conclusion sections.

Focus on Clarity and Brevity.

With these revisions addressed, the paper will be greatly improved and may be suitable for acceptance. Based on the given information, there is no concern about double publication/research ethics/publication ethics.

PLOS authors have the option to publish the peer review history of their article (what does this mean? ). If published, this will include your full peer review and any attached files.

**Do you want your identity to be public for this peer review?** For information about this choice, including consent withdrawal, please see our Privacy Policy .

Reviewer #1: No

Reviewer #2: No

Reviewer #3: **Yes: ** Wojciech Iwaniak

**Figure resubmission:**

**Reproducibility:**



---

## [Decision Letter · Decision Letter 1]

Dear Dr Mumba,

We are pleased to inform you that your manuscript 'Anthrax, Livelihood Vulnerability, and Food Insecurity in Selected Game Management Areas in Zambia: A Mixed-Methods Analysis at the Human-Wildlife-Livestock Interface' has been provisionally accepted for publication in PLOS Neglected Tropical Diseases.

Best regards,

Georgios Pappas

Section Editor

Georgios Pappas

Section Editor

Shaden Kamhawi

co-Editor-in-Chief

Paul Brindley

co-Editor-in-Chief

-

Reviewer's Responses to Questions

**Key Review Criteria Required for Acceptance?**

**Methods**

-Are the objectives of the study clearly articulated with a clear testable hypothesis stated?

-Is the study design appropriate to address the stated objectives?

-Is the population clearly described and appropriate for the hypothesis being tested?

-Is the sample size sufficient to ensure adequate power to address the hypothesis being tested?

-Were correct statistical analysis used to support conclusions?

-Are there concerns about ethical or regulatory requirements being met?

Reviewer #2: I am happy with the revised version and believe that the methods are now adequately described and appropriate for the study.

Reviewer #3: Objectives and Hypothesis: The objectives of the study are adequately stated in the abstract and introduction, seeking to understand the socio-economic impacts of anthrax outbreaks, associated stressors, food insecurity, and livelihood vulnerabilities among rural Zambian communities. While no formal, explicit testable hypotheses are presented, the study explores the possible relationships between anthrax outbreaks, livelihood strategies, and food security.

Study Design: The mixed-methods approach, using quantitative surveys and qualitative focus group discussions simultaneously, is appropriate considering the complexity and heterogeneity of factors affecting the socio-economic burden of anthrax.

Population Description: The population definition is specific, focusing on rural communities in designated Zambian Game Management Areas (GMAs). This choice is appropriate, relying on the frequency of anthrax and the interactions of humans, wildlife, and livestock in these regions.

Sample Size: The paper provides a sample size calculation for the quantitative survey. The rationale for the qualitative sample size (6 focus groups) is included in relation to data saturation.

Statistical Analysis: The authors claim appropriate statistical analyses were used to screen the quantitative data by performing chi-squared tests for associations between categorical variables. Details of approaches to qualitative data analysis, such as thematic analysis based on focus groups, are provided.

Ethical and Regulatory Requirements: The study was ethically approved by ERES Converge, and anonymization processes are described in detail to demonstrate compliance with ethical standards. These ensure the anonymity of all participants and state the right to leave the study at any time.

**Results**

-Does the analysis presented match the analysis plan?

-Are the results clearly and completely presented?

-Are the figures (Tables, Images) of sufficient quality for clarity?

Reviewer #2: I am happy with the revised version and believe that the results are now adequately described.

Reviewer #3: Match with Analysis Plan: Text in the document indicates that the analysis performed matched that of the analysis plan. Qualitative data were analyzed using thematic analysis, which is described in the Methods section. Quantitative data were analyzed using suitable statistical tests (chi-squared tests referred to in the text).

Clarity and Completeness: Results are clearly presented, with findings primarily presented as quantitative or qualitative. Main findings are summarized, and tables are referenced to describe socio-demographic data, but they are not complete tables, only parts of them. Illustrative quotations also represent the qualitative results well.

Figures and Tables Quality: Only parts of tables 1 and 2 were visible and difficult to assess. More detail to be able to display the clarity of any visual aids.

**Conclusions**

-Are the conclusions supported by the data presented?

-Are the limitations of analysis clearly described?

-Do the authors discuss how these data can be helpful to advance our understanding of the topic under study?

-Is public health relevance addressed?

Reviewer #2: yes, all these criteria are fulfilled

Reviewer #3: Support by Data: The conclusions are supported by the presented data.

Limitations Described: The limitations are not explicitly stated in the conclusion section.

Advancement of Understanding: The results will provide insights for this particular study topic.

Public Health Relevance: The public health relevance is well articulated, indicating the importance of interventions for controlling anthrax outbreaks, diversifying livelihood, and increasing food security. The study emphasizes the importance of the risks of zoonotic diseases in rural settings and reminds us that advice cannot simply be given but rather should be community-based, including empowerment among community members, especially women.

**Editorial and Data Presentation Modifications?**

Reviewer #2: There are a few proofing errors:

Line 158: I suggest changing from '...makes understanding of disease dynamics better.' to '...facilitates understanding of disease dynamics.'

Line 465: I suggest adding 'to' (...often leads to animals being moved...)

Line 481: I suggest adding 'spores' (...expose dormant anthrax spores and make them...)

Line 482: remove 's' (soil condition)

Line 285: change form is to are (..bacteria are present...)

Reviewer #3: No comments

**Summary and General Comments**

Reviewer #2: I am content with the revised version of the manuscript

Reviewer #3: No comments

PLOS authors have the option to publish the peer review history of their article (what does this mean? ). If published, this will include your full peer review and any attached files.

**Do you want your identity to be public for this peer review?** For information about this choice, including consent withdrawal, please see our Privacy Policy .

Reviewer #2: **Yes: ** Susanna Sternberg Lewerin

Reviewer #3: **Yes: ** Wojciech Iwaniak

---

## [Editor Report · Acceptance letter]

Dear Dr Mumba,

We are delighted to inform you that your manuscript, "Anthrax, Livelihood Vulnerability, and Food Insecurity in Selected Game Management Areas in Zambia: A Mixed-Methods Analysis at the Human-Wildlife-Livestock Interface," has been formally accepted for publication in PLOS Neglected Tropical Diseases.

Best regards,

Shaden Kamhawi

co-Editor-in-Chief

Paul Brindley

co-Editor-in-Chief
